# Meta-Analysis of Effects of Melatonin Treatment on Plant Drought Stress Alleviation

**Yuzhe Wang** [1], **Siyu Gun** [2,3], **Yaoyu Li** [1] **and Laiye Qu** [2,3,*]

1    Faculty of Environment and Life, Beijing University of Technology, Beijing 100124, China
2    State Key Laboratory of Urban and Regional Ecology, Research Center for Eco-Environmental Sciences, Beijing 100085, China
3    Research Center for Eco-Environmental Science, University of Chinese Academy of Sciences, Beijing 100049, China
*    Correspondence: lyqu@rcees.ac.cn

**Abstract:** Due to the increasing frequency of extreme drought events worldwide in recent years, improving the adaptability of plants to arid environments has become an important research topic. In particular, many studies have investigated the effects of melatonin on drought stress mitigation in plants. However, most of these studies were limited in terms of the number of sampling sites or regional scale, and thus we lack a comprehensive understanding of the effects of the exogenous application of melatonin on drought stress mitigation in plants on a global scale. In this study, we integrated previous research into the physiological and growth effects of melatonin application in arid environments worldwide and analyzed the response of plants to different melatonin concentrations, application methods, and different drought degrees in order to provide a scientific basis for promoting the use of melatonin in alleviating plant drought stress. The data used in this study were obtained from the "Web of Science" database, where the keywords "drought & melatonin" were used to search the relevant literature. In total, 61 valid publications with 140 data sets were retrieved. A meta-analysis was performed using the data with no melatonin treatment as the control group and melatonin treatment as the experimental group. Melatonin application significantly increased the plant biomass, chlorophyll content, and antioxidant enzyme activity to alleviate the damage caused by drought stress. The accumulated biomass and accumulation of chlorophyll in plants varied with the melatonin concentration. The threshold value range was identified as 80–120 $\mu$mol L$^{-1}$, and the effect of melatonin on the accumulation of biomass and chlorophyll decreased gradually above this range. In addition, the effects of various spraying methods on the mitigation of drought stress in plants differed significantly. Soil application had greater effects on reactive oxygen species scavengers in plants than foliar spraying. Moreover, the plant leaf membrane lipid peroxidation degree was relatively low, and the plant body chlorophyll content was higher under soil application than foliar spraying, and the cumulative biomass was lower than that with foliar spraying. The effects of melatonin on mitigating plant drought stress also varied under different drought levels when using the same melatonin concentration and application method. Soil irrigation is most effective if the main aim is to improve plant stress resistance and the below-ground root biomass, but foliar spraying is most effective for increasing photosynthesis and plant biomass.

**Keywords:** drought stress; melatonin; meta-analysis; plant growth

## 1. Introduction

Due to accelerated global warming and climate change, the frequency of extreme weather events is rapidly increasing [1]. In particular, the increased frequency of drought events due to climate change is a global issue that must be addressed [2]. According to the United Nations Intergovernmental Panel on Climate Change (IPCC), if global warming reaches 1.5 °C compared with the pre-industrial revolution era, the frequency of global

droughts will be twice as high as before the industrial revolution [3]. This report also suggests that between 2013 and 2016, the land area threatened by drought increased at a rate of 1% per year, with more than 500 million people living in areas affected by drought and desertification in 2015 [3]. Increasingly severe droughts have adversely affected people's lives and livelihoods, and they are among the most costly natural disasters. In addition to increasing the risk of forest fires, drought leads to a decline in biodiversity and causes other adverse consequences, such as a strain on urban water supplies, although reduced crop yields are the most direct effect of drought on humans. Expansion pressure, nutrient concentrations, and assimilated carbon substances required for plant growth are reduced under drought, and thus the rate of nutrient uptake by plants is lower under such conditions [4,5]. Studies have shown that differences in plant height, biomass, leaf size, and stem thickness may also lead to reduced photosynthetic rates and imbalances in the distribution of the limited nutrients absorbed by plants, thereby hindering the growth of crops [6,7].

Melatonin (MT), also known as pinealectin, has been detected in almost all plants and plant products since it was first detected in 1995 [8,9]. Studies have shown that both self-synthesized and exogenous MT can enhance the adaptation of plants to various stresses [9]. The application of MT in agricultural production can potentially increase food production, where its effect may be related to MT acting as an antioxidant to protect chlorophyll. Therefore, leaf senescence is delayed, photosynthetic efficiency is increased, and the adaptability of plants to various stresses is enhanced. The exogenous application of MT is often used to alleviate drought stress in plants, and this approach has been promoted and applied in many studies. For example, Dai et al. (2020) investigated different varieties of oilseed rape (*Brassica napus* L.) and showed that exogenous MT application enhanced plant drought resistance by altering the stomatal activity, root growth, and peroxidase (POD) gene expression [10]. Hossain et al. (2020) studied buckwheat (*Fagopyrum tataricum* L.) and found that MT could prevent reactive oxygen species (ROS)-induced damage to plants by enhancing the expression of antioxidant enzymes as well as other enzymes [11]. Sadak et al. (2020) investigated two varieties of flax (*Linum usitatissimum* L.) and found that MT in plants could directly scavenge ROS to alleviate the plant damage caused by drought stress [12].

Exogenous MT has been applied to plants under different drought conditions, but it is not clear whether different MT concentrations and spraying methods might affect the alleviation of drought stress and the extent of any differences. Therefore, in this study, we integrated previous data obtained in related research into MT application for drought stress alleviation in plants and conducted a meta-analysis to synthesize the results in order to elucidate the effects of different exogenous MT application concentrations, application methods, and drought stress levels on plant growth, physiology, and antioxidant enzymes. The findings obtained in this study have significant implications for the application of MT in agricultural production to combat drought, improve crop yields, prevent land desertification, and ensure food security.

## 2. Materials and Methods

### 2.1. Database Set Up

A database search was conducted on 16 April 2022 using the Web of Science (www.webofscience.com) platform for the subject: drought & melatonin.

Literature type: ARTICLE

Time span: all years

In total, 243 publications were retrieved, which were considered valid when they satisfied the following conditions: (1) treatment performed using exogenous MT with a blank drought treatment; (2) the number of treatment replicates and standard deviations in the experiment were clearly presented; (3) the experiment was performed in a controlled environment, laboratory, growth chamber, or greenhouse environment, and (4) data with repeated measurements were available, and only the last result was selected.

After screening, 61 valid publications were retained. Observations of any one plant species, soil water content, exogenous MT applied concentration, and incubation time from one study were pooled after analysis and treated as independent samples. The mean (treatment mean and control mean), standard deviation (treatment standard deviation and control standard deviation), and number of treatment replicates (number of treatment replicates and number of control replicates) were extracted for each index in the study. Studies that presented standard errors were retained after performing conversion to the standard deviation ($SD = SE \times \sqrt{N}$). In order to directly extract data from tables and figures, images were extracted using Web Plot Digitizer software. In total, 170 study cases were imported and compiled in R version 4.2.0.

### 2.2. Exogenous MT Treatment Concentration

In order to analyze the effects of different concentrations of exogenous MT, we set the treatment without the addition of exogenous MT in a study as the control group and the treatment with exogenous MT application as the experimental group. In addition, in order to facilitate the analysis and comparisons of the data, the exogenous MT concentrations in the experimental group were grouped into three classes according to the criteria in Table 1.

**Table 1.** Groups of melatonin concentrations and the sample sizes.

| | Indices | Melatonin Concentration/($\mu mol \cdot L^{-1}$) | | |
| --- | --- | --- | --- | --- |
| | | Low Concentration (0,60] | Medium Concentration (60,120] | High Concentration (120,200) |
| Sample size | Above-ground biomass | 9 | 8 | 2 |
| | Underground biomass | 19 | 18 | 10 |
| | Total biomass | 20 | 23 | 9 |
| | Plant height | 20 | 24 | 12 |
| | Root diameter | 9 | 8 | 2 |
| | Chlorophyll a | 11 | 16 | 11 |
| | Chlorophyll b | 11 | 16 | 11 |
| | Total chlorophyll | 18 | 29 | 19 |
| | $H_2O_2$ | 34 | 49 | 32 |
| | MDA | 31 | 42 | 36 |
| | POD | 23 | 34 | 22 |
| | SOD | 30 | 45 | 36 |
| | CAT | 34 | 46 | 34 |
| | APX | 14 | 21 | 16 |

$H_2O_2$: hydrogen peroxide; MDA: malondialdehyde; POD: peroxidase; SOD: superoxide dismutase; CAT: catalase; APX: ascorbate peroxidase.

### 2.3. Drought Level

To analyze the effects of different drought levels, plants treated with drought alone and without exogenous MT application in each study were set as the control group, and plants treated with exogenous MT application under the same drought level were set as the experimental group. The number of plants treated with exogenous MT application by soil irrigation was low in previous studies, and they generally applied MT at between 30–50% of the field water content. Thus, in order to detect significant differences in the plant drought stress mitigation effect under different drought stress levels with the same exogenous MT concentration and the same exogenous MT application method, a field water content range of 30–40% was classified as level I drought and a field water content range of 40–50% was classified as level II drought, as shown in Table 2.

**Table 2.** Groups of drought levels and sample sizes.

| Drought Degree | Field Capacity(%) | Sample Size | | |
|---|---|---|---|---|
| | | $H_2O_2$ | CAT | APX |
| level I drought | (30, 40] | 12 | 15 | 15 |
| level II drought | (40, 50] | 25 | 24 | 15 |

$H_2O_2$: hydrogen peroxide; CAT: catalase; APX: ascorbate peroxidase.

### 2.4. Mechanism of Melatonin Regulating Drought Stress

Melatonin protects the photosynthetic apparatus from the deleterious effects of drought by reducing chlorophyll degradation and recovering the chloroplast structure resulting in the recovery of the photosynthetic efficiency of plants [13,14]. Additionally, melatonin protects plants from the negative effects of drought-induced oxidative stress by enhancing the antioxidant system and increasing osmolytes in plants [13,15]. In addition, melatonin protects plants from the negative effects of drought by regulating mitogen-activated protein kinase (MAPK) signaling [16]. The expression of stress-protective proteins increased by MAPK enhanced transcriptional regulation and translational regulation [17]. In conclusion, melatonin regulates photosynthetic response, oxidative stress, antioxidative defense system, and other biological processes to increase the drought resistance of plants [17].

### 2.5. Meta-Analysis

In order to eliminate the effects of different laboratories, different measurement environments, and different plant growth environments on plants, we performed a meta-analysis based on all of the extracted data using a random-effect model [18] with RStudio software by considering both within-case differences in effect values and between-case variation. The effect value (E) was calculated for the data as the effect ratio (R) between the experimental treatment and control to calculate the effect value (M value) [19,20] as follows:

$$R = \frac{X_e}{X_c}$$

$$M = \ln R = \ln\left(\frac{X_e}{X_c}\right)$$

where $X_c$ is the mean of the corresponding indicators in the control group and $X_c$ is the mean of the corresponding indicators in the treatment group. A negative effect value indicates a decreased index in response to the application of melatonin, whereas positive values indicate an increase.

In order to identify differences in temperature on species of plants, we use Residual maximum likelihood (REML) to calculate the variance between cases. The heterogeneity (QB) between data were assessed when comparing different categories of effect values with different variance between cases, and if QB was statistically significant (i.e., $p < 0.05$; note that $p < 0.05$ is omitted in the following figures), the data were subdivided according to the classification criteria given above. The 95% confidence interval was obtained by the self-help method [21]. When 95% of the indicators in a category confidence interval did not overlap with 0, it was considered that exogenous MT treatment caused a significant change in the indicator. When the 95% confidence intervals of the same indicator under different classifications did not overlap, significant differences were indicated in different classifications.

## 3. Results

### 3.1. Effects of Different MT Concentrations on Plants

3.1.1. Effects of Different MT Concentrations on Plant Biomass

The accumulated biomass and accumulation of chlorophyll in plants varied with the MT concentration, where the threshold value range was 80–120 µmol $L^{-1}$. When the MT concentration exceeded this threshold range, the effect of MT on increases in the accumulation of biomass and chlorophyll gradually decreased. Regression analysis revealed that the plant height, root diameter, below-ground biomass, and total biomass all tended to increase and then decrease as the MT concentration increased (Figures 1 and 2), and the maximum effect value occurred in the concentration range of 80–120 µmol $L^{-1}$. When the concentration was greater than 120 µmol $L^{-1}$, the effect value gradually decreased as the concentration increased.

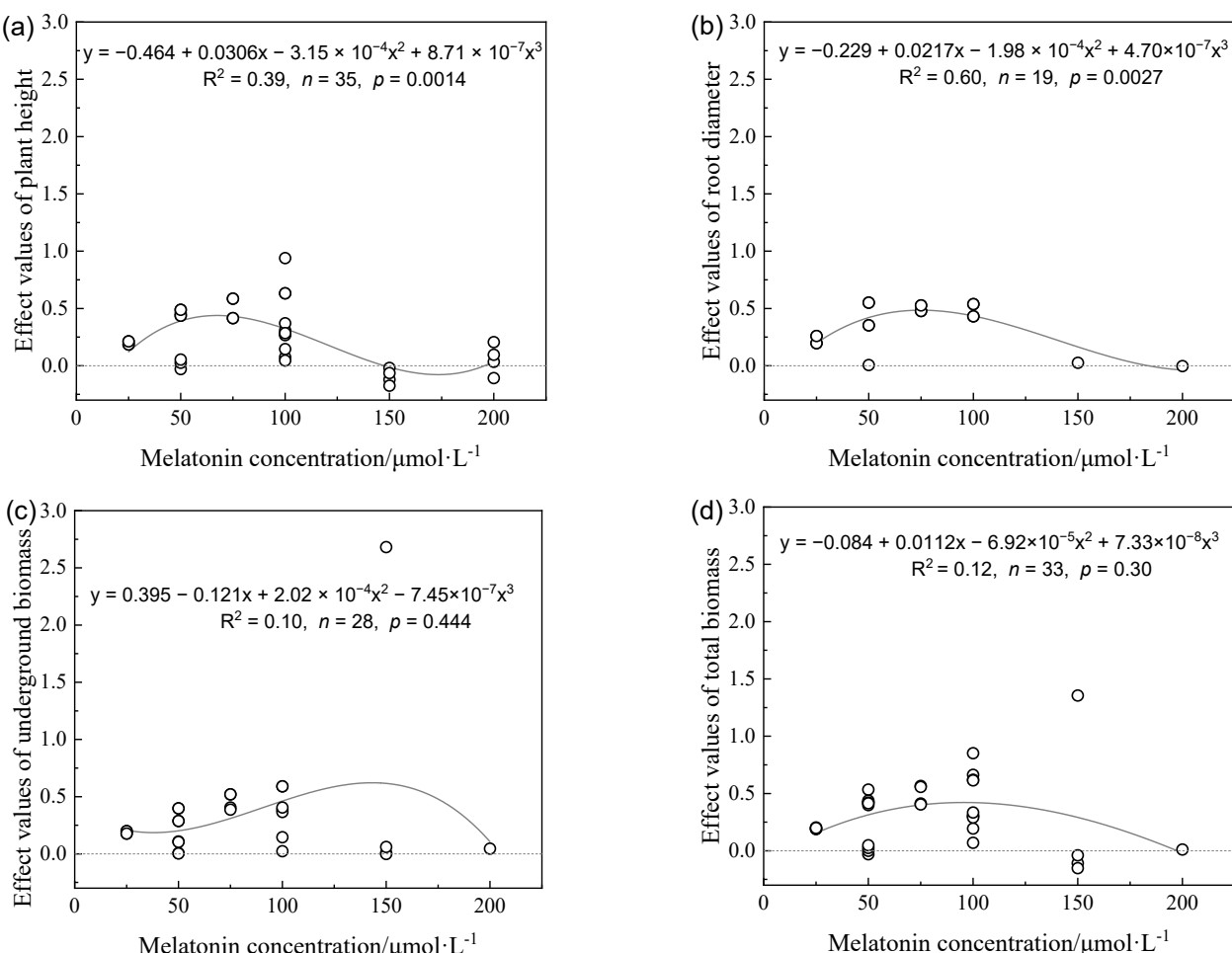

**Figure 1.** Effects values of plant height (**a**), root diameter (**b**), underground biomass (**c**), and total biomass (**d**) in different melatonin concentrations on alleviating plant growth inhibition under drought stress.

3.1.2. Effects of Different MT Concentrations on Photosynthetic Pigments in Plants

The changes in chlorophyll a and b and the total chlorophyll effect index under MT application were similar to those in the biomass and rootstock, plant height, and other indicators. The maximum values of chlorophyll a and b and the total chlorophyll index occurred in the applied exogenous MT concentration range of 65–80 µmol $L^{-1}$. Subsequently, the effect on the chlorophyll content decreased as the MT concentration increased (Figure 2).

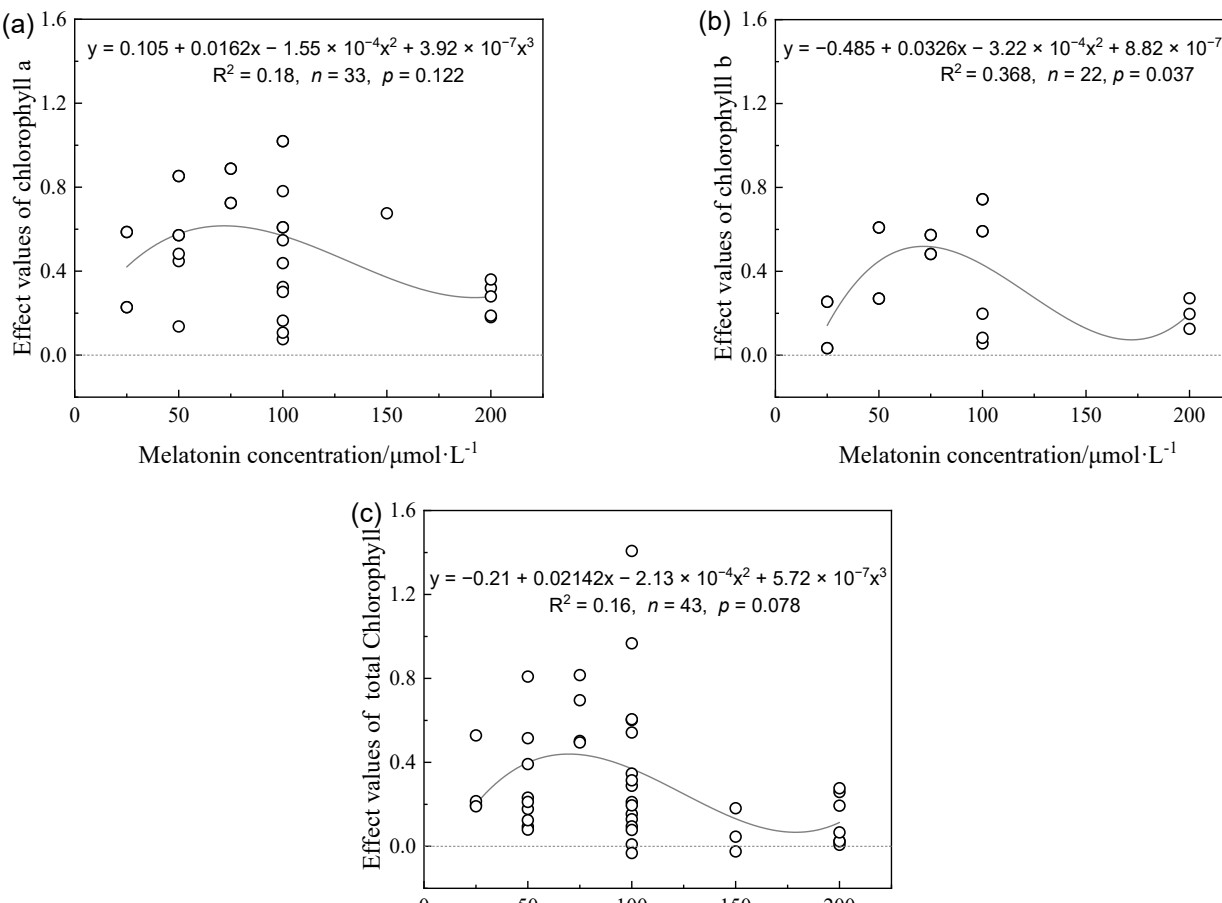

**Figure 2.** Effects values of Chlorophyll a (**a**), Chlorophyll b (**b**), and total chlorophyll (**c**) in different melatonin concentrations on alleviating plant growth inhibition under drought stress.

### 3.1.3. Effects of Different MT Concentrations on Antioxidant Properties of Plants

The $H_2O_2$ contents of plants tended to decrease at MT concentrations in the range of 0–100 μmol $L^{-1}$ and gradually increased from 100–200 μmol $L^{-1}$ (Figure 3a). The malondialdehyde (MDA) contents of plants decreased as the MT concentration increased and gradually stabilized (Figure 3b). Compared with the drought plant control, low and medium MT concentrations (0–120 μmol $L^{-1}$) caused significant increases in the plant superoxide dismutase (SOD) and POD contents, whereas high MT concentrations (120–200 μmol $L^{-1}$) caused decreases in the plant SOD and POD contents (Figure 4a,b). The effect values for catalase (CAT) increased initially as the MT concentration increased to reach a maximum in the range of 100–150 μmol $L^{-1}$, but with no further increases (Figure 4c). MT concentrations below 60 μmol $L^{-1}$ had no significant effects on the ascorbate peroxidase (APX) contents of plants. The APX contents gradually increased and stabilized when the MT concentration was greater than 60 μmol $L^{-1}$.

### 3.2. Plant Responses to Different MT Application Methods

Meta-analysis results showed that the responses of plants to foliar spraying with different MT concentrations were basically the same as those under soil irrigation. Under the same exogenous MT concentration, the effects on the chlorophyll a, chlorophyll b, total chlorophyll, $H_2O_2$, MDA, APX, CAT, and POD contents were significantly better in soil-irrigated plants than in foliar sprayed plants (Figure 5). However, foliar spraying had greater effects than soil irrigation on the root diameter, plant height, above-ground biomass, below-ground biomass, and total biomass.

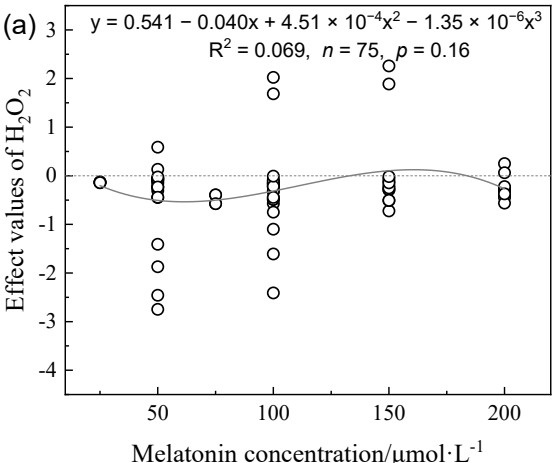

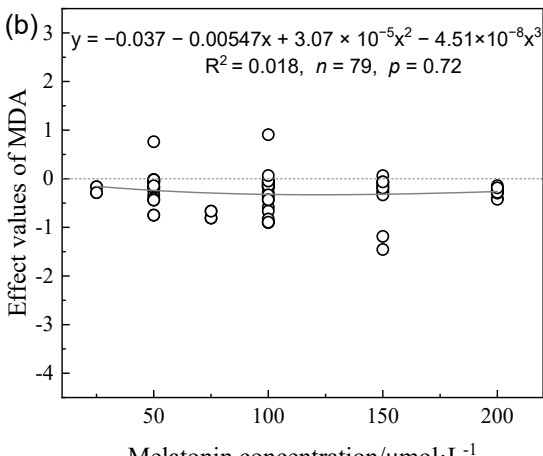

**Figure 3.** Effects values of $H_2O_2$ content (**a**) and MDA level (**b**) in different melatonin concentrations on alleviating plant growth inhibition under drought stress.

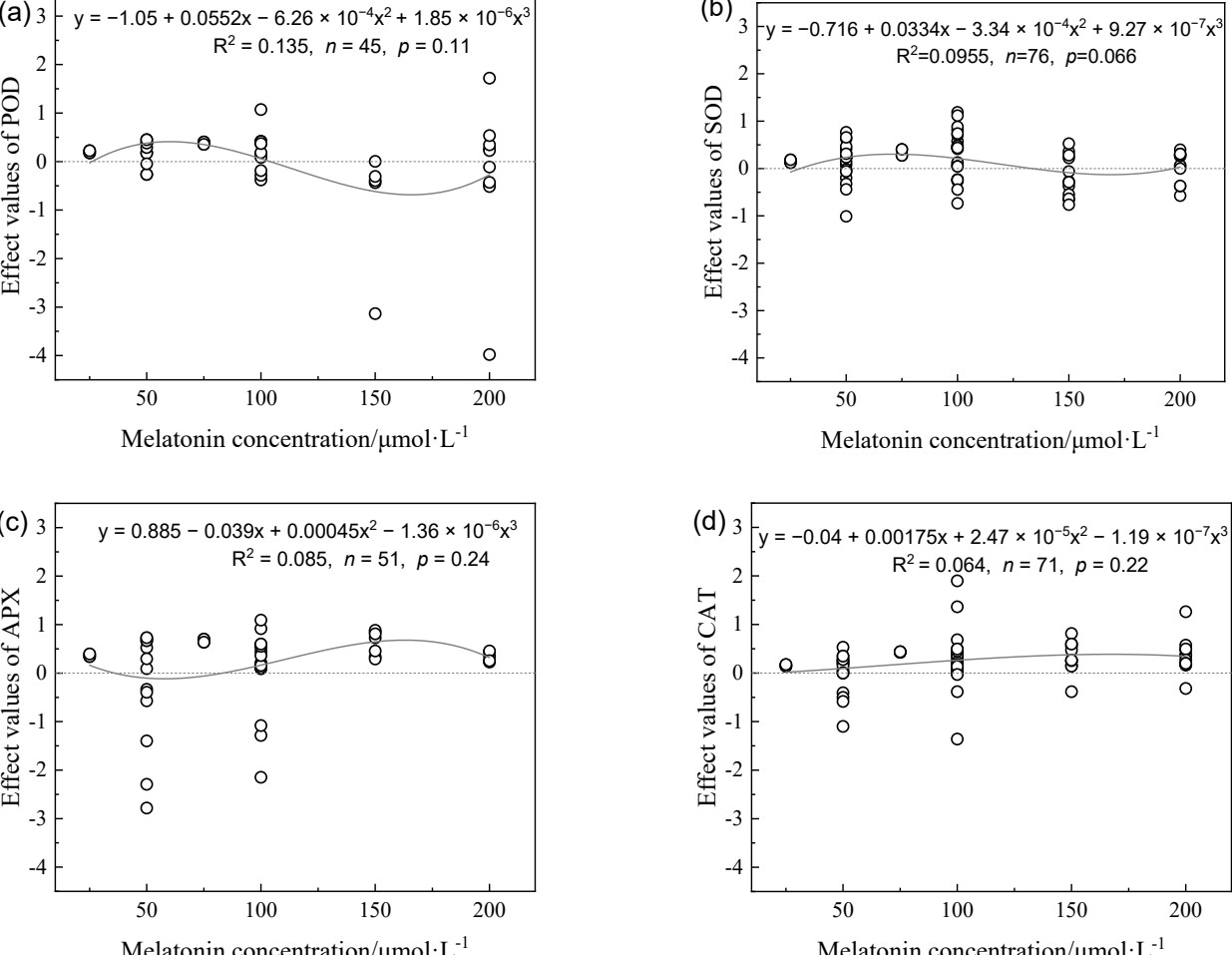

**Figure 4.** Effects values of POD activity (**a**), SOD activity (**b**), APX activity (**c**), and CAT activity (**d**) in different melatonin concentrations on alleviating plant growth inhibition under drought stress.

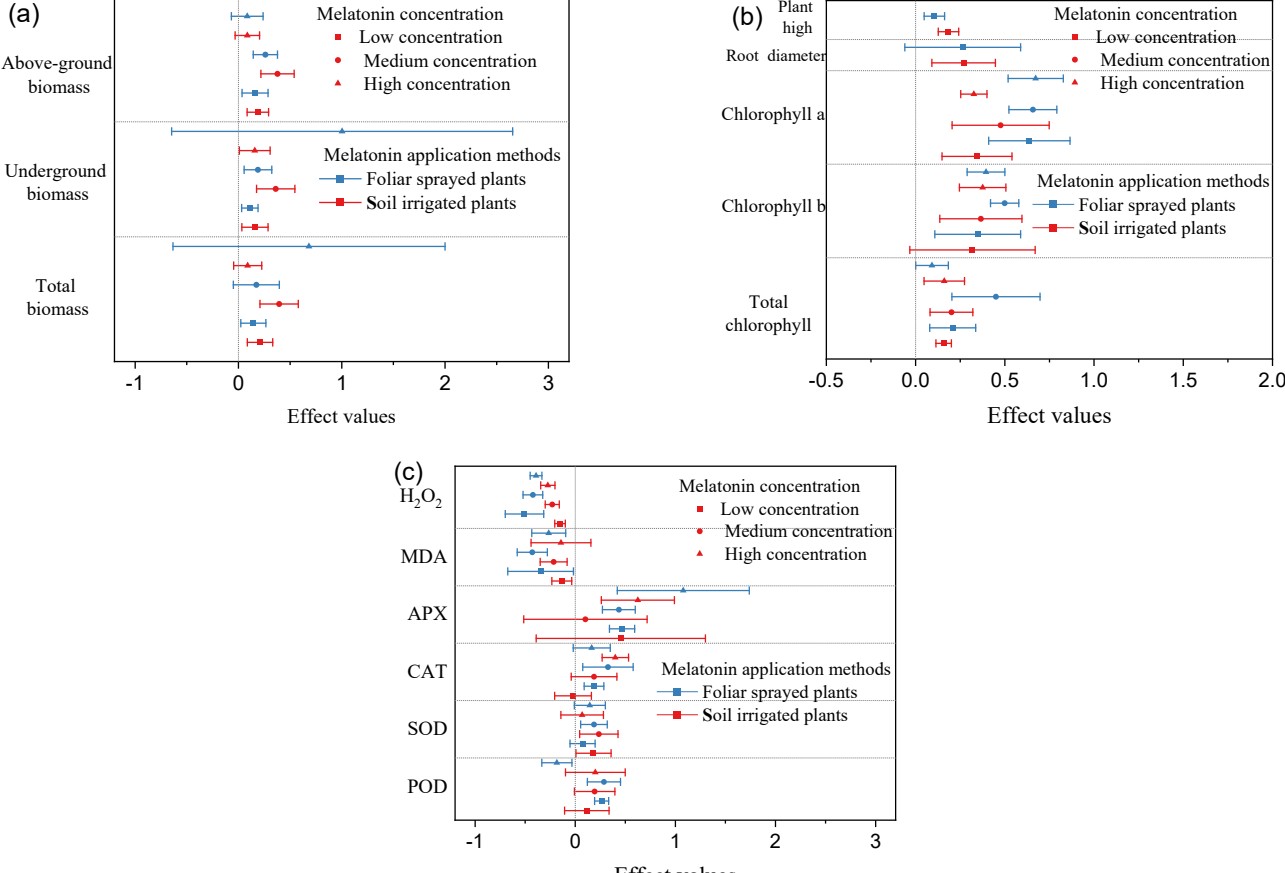

**Figure 5.** Effect value analysis based on different melatonin concentrations and application methods. Symbols represent the mean percent change in drought environment relative to drought environment with melatonin infliction, and the bars show the 95% bootstrapped confidence intervals. (**a**) Plant biomass; (**b**) Growth factors; (**c**) Membrane lipid peroxidation and antioxidant enzyme activities.

### *3.3. Plant Responses to Different MT Concentrations and Application Methods under Different Drought Stress Levels*

Plants were affected by different drought levels under the same exogenous MT concentration and application method, where significant differences were found in each index (Figure 6). Under foliar spraying, the application of a medium concentration of exogenous MT reduced the $H_2O_2$ effect values for plants under level I drought by about 50% compared with level II drought and increased the CAT and APX effect values for plants under level I drought by about 54% and 91%, respectively, compared with level II drought. The application of a high concentration of exogenous MT had no significant effect on the effect values for CAT in plants under level II drought, but the effect was significant under level I drought. Under soil irrigation, the application of a medium concentration of exogenous MT increased the effect values for $H_2O_2$ in plants under class I drought by about 5% compared with those under level II drought, and the effect values for CAT and APX in plants under level I drought increased by about 47% and 194%, respectively, compared with those under level II drought. The effect values for APX and CAT in plants under drought level II were not significant after applying high concentrations of exogenous MT, but the effects were significant for both under drought level I conditions.

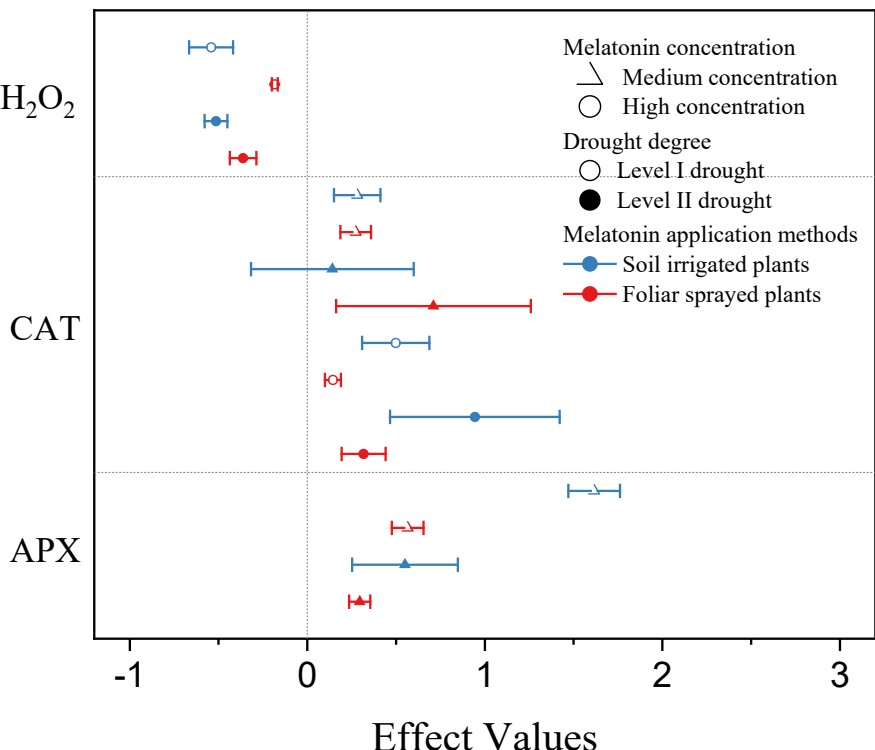

**Figure 6.** Effect sizes for plants under different melatonin application methods and concentrations and drought levels. Symbols represent the mean percent change in drought environment relative to drought environment with melatonin infliction, and the bars show the 95% bootstrapped confidence intervals.

## 4. Discussion

### 4.1. Effects of Different Exogenous MT Concentrations on Plant Growth

Under drought stress, the chlorophyll contents of plants decrease to reduce the photosynthetic capacity [22], thereby leading to excess excitation energy in plants and redox imbalance, which reduces the accumulated plant biomass [23,24]. Exogenous MT is a plant growth regulator that controls various physiological and biochemical indicators in plants, and thus it can alleviate the effects of biotic and abiotic stresses on plants. Numerous studies have shown that MT plays important roles in regulating the growth and development of vascular plants, and it acts as an antioxidant to protect plants from biotic stresses [10]. MT has antioxidant effects on several plant species, such as rice [22], maize [23], apple [24], and grape [25]. Li [26] found that the exogenous application of MT to *Begonia lappa* reduced ROS-induced oxidative damage through the direct scavenging of hydrogen peroxide ($H_2O_2$) and by enhancing the antioxidant enzyme activity. MT application also enhanced the tolerance to salt and drought stresses in soybean by upregulating the expression of genes repressed by salt stress [27]. MT can alter many plant characteristics, including germination, seedling growth, flowering time, grain yield, and senescence [28,29]. Another unique function of MT in plants is its growth hormone activity, which promotes plant growth.

MT is a plant growth regulator that can regulate various physiological and biochemical indicators in plants to alleviate the effects of biotic and abiotic stresses. The results of the meta-analysis conducted in the present study showed that the above-ground biomass, below-ground biomass, and total biomass were greater in plants under drought stress than those without MT, thereby indicating that exogenous MT can effectively alleviate the inhibitory effects of drought stress on plant growth. Moreover, the biomass tended to increase initially and then decrease as the MT concentration increased, where the maximum effect value occurred in the range of 80–120 μmol L$^{-1}$. However, the effect values for growth did not increase with the MT concentration and they tended to decrease, as also

shown in many previous studies [30,31]. Therefore, the application of moderate MT concentrations was most beneficial in alleviating drought stress effects in plants. Chlorophyll is an important photosynthetic pigment in plants with important roles in light absorption and transport [32]. In this study, we found that the application of a moderate amount of MT under drought conditions increased the photosynthetic pigment contents of plants, as also shown in previous studies [33], probably because MT can effectively slow down the metabolic decomposition of chlorophyll and maintain the chloroplast ultrastructure to increase the accumulation of chlorophyll in plants. In addition, the relationship between the MT concentration and photosynthetic pigment content was basically the same as that with the biomass, which suggests that exogenous MT application can enhance photosynthesis by increasing the photosynthetic pigment contents of plants to increase the biomass and alleviate the effects of drought. Drought stress leads to the accumulation of large amounts of ROS ($O_2$ and $H_2O_2$) in plants to damage the plant biofilm system (increased MDA content), and the levels of reactive oxygen scavengers (SOD, POD, CAT, and APX) must increase significantly to remove the excess ROS [34]. The meta-analysis conducted in this study showed that exogenous MT application at medium concentrations under drought conditions resulted in the highest antioxidant enzyme activities (SOD, POD, CAT, and APX) in plants. Thus, exogenous MT application can regulate the antioxidant system in plants to allow them to adapt to drought conditions. In addition, MT is considered a broad-spectrum antioxidant and free radical scavenger [35] that may directly eliminate the excess ROS produced by drought stress.

### 4.2. Effects of Different MT Application Methods on Plant Growth

Plants respond to water deficit through different mechanisms, including physiological, metabolic, and defense systems. These responses include a well-developed root system, changes in phytohormones, enhanced antioxidant enzyme systems, stomatal closure, and the production of low molecular osmolytes [10]. Most studies of MT-enhanced stress tolerance considered leaf responses [22,23], but they also focused on the responses of the roots to MT. In particular, Zhang et al. [28] investigated the effect of MT on root growth in cucumber and canola. Chen et al. [36] found that MT application promoted root growth in mustard-type canola, and Sarropoulou et al. [37] reported the promotion of adventitious root regeneration in cherry stem tip explants. Thus, MT seems to play an important role in abiotic stress resistance, especially drought and salt stress. Dai et al. [10] found that MT promoted the growth of lateral and primary roots but also enhanced the activity of CAT in the primary root tips. In addition, MT promoted stomatal opening (i.e., increased stomatal aperture) to enhance gas exchange in the leaves of drought-sensitive and tolerant genotypes. MT was applied to different plant parts using various methods, which can mainly be classified as foliar spraying and soil irrigation. Our meta-analysis showed that the activity of ROS removers was generally higher in plants treated by soil irrigation than foliar spraying, thereby indicating that the antioxidant enzyme system was relatively more effective at removing ROS in the former, and the lower $H_2O_2$ and MDA contents also confirmed that the antioxidant enzyme systems were stronger in soil irrigated plants. In addition, the chlorophyll a, chlorophyll b, and total chlorophyll contents were higher in soil irrigated plants than foliar sprayed plants, although the chlorophyll contents of plants grown under the normal soil field water holding capacity did not recover [12], and thus plants became more capable of performing photosynthesis and produced more organic matter [38].

Respiration in higher plants includes the cytochrome respiration pathway (COX pathway) and the alternate respiration pathway (AOX pathway). The total plant respiration increases under drought stress, with a slight increase in respiration via the COX pathway and a significant increase in respiration via the AOX pathway [39], and AOX can effectively reduce ROS production [40]. In addition, the increase in ROS in plants inhibits D1 protein turnover, thereby leading to a decrease in respiration via the AOX pathway [41]. After the application of exogenous MT, the ROS content decreases in plants to enhance D1

protein turnover, and respiration via the AOX pathway increases while respiration via the COX pathway remains largely unchanged, and thus the total respiration increases. Our meta-analysis showed that the plant biomass and growth parameters were lower under the application of exogenous MT by soil irrigation than foliar spraying, and thus the net photosynthetic rate decreased the increase in the respiration rate more than the increase in the photosynthetic rate. Therefore, we hypothesize that the increase in respiration was greater than the increase in photosynthesis in soil-irrigated exogenous MT plants compared with foliar sprayed plants, thereby resulting in a lower net photosynthetic capacity compared with foliar sprayed plants, so the above-ground biomass, underground biomass, total biomass, plant height, and root diameter were lower than those in foliar sprayed plants.

In addition, we explored whether MT mitigated the effects of plant drought stress under different drought stress levels using various irrigation methods. The plant stress mitigation effect of MT was greater under drought level I than drought level II when using the same irrigation method. However, due to the small sample size, it was not possible to analyze all of the indicators, although some of the indicators suggested that the drought stress mitigation effect of MT differed significantly under different drought levels.

## 5. Conclusions

In this study, we conducted a meta-analysis based on 61 publications with 140 data sets and showed that the application of exogenous MT could alleviate the effects of drought stress on plants. The accumulated biomass and photosynthetic pigment contents of plants tended to increase initially and then decrease as the MT concentration increased, and the effect value for exogenous MT application on drought stress mitigation was highest when the exogenous MT concentration range was 80–120 $\mu$mol L$^{-1}$. The addition of exogenous MT mainly increased the activity of reactive oxygen scavengers and the contents of photosynthetic pigments in plants to reduce the degree of membrane lipid peroxidation in plant leaves, as well as enhancing the photosynthetic rate to alleviate the effects of drought stress. The methods used to apply exogenous MT, comprising foliar spraying method and soil irrigation, also influenced the effects of MT on plant drought stress mitigation. Soil irrigation is most effective if the main aim is to improve plant stress resistance and the below-ground root biomass, but foliar spraying is most effective for increasing photosynthesis and plant biomass. In arid areas, the roots of plants often reach deep underground, and the effect of soil irrigation might not be optimal. Therefore, we suggest that applying MT at a concentration of 100 $\mu$mol L$^{-1}$ by foliar spraying in arid regions may provide greater relief from drought stress. The results obtained in this study provide the basis for mitigating the effects of plant drought stress in the future global drought context with MT application.

**Author Contributions:** Conceptualization, Y.W., S.G., Y.L. and L.Q.; formal analysis, Y.W.; investigation, Y.L.; methodology, Y.W.; supervision, L.Q.; writing—original draft, Y.W.; writing—review and editing, Y.W., S.G., Y.L. and L.Q. All authors have read and agreed to the published version of the manuscript.

**Funding:** This research was funded by The National Key Research and Development Program of China, grant number 2017YFE0127700.

**Institutional Review Board Statement:** Not applicable.

**Acknowledgments:** The authors would like to thank all teachers and student assistants for their invaluable support.

**Conflicts of Interest:** The authors declare no conflict of interest.

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
