# Peer review of "Meta-Analysis of Effects of Melatonin Treatment on Plant Drought Stress Alleviation"

_agriculture, doi:10.3390/agriculture12091335_

Round 1

Reviewer 1 Report

The authors need to address some minor modifications. Below are some specific comments:

For minor correction:

1)      There should not be any confusing recommendation in the manuscript. Need to check the Abstract (Line: 33-35) and Conclusion (Line: 355-357), and make it clear.

2)      Need to check and set proper punctuation in the sentence “The application of MT .................. to various stresses.”

3)      In the Fig.1 (a), the words ‘plant high’ should be corrected as ‘plant height’.

4)      In the Fig.2 (c), the words ‘chlorophyll total’ should be corrected as ‘total chlorophyll’.

5)      In the Fig.5 (b), the words ‘plant high’ and ‘chlorophyll total’ should be corrected as ‘plant height’ and ‘total chlorophyll’, respectively.

Reviewer 2 Report

1- Since the work is theoretical research it is expected to have a larger data base

2- clear and adequate description of methods is required

details of MT action mechanism must be added in molecular level

3- English can be improved

4- Figure legends must be more informative.

Reviewer 3 Report

This is an intriguing study, and the authors compiled a dataset from many published research papers. The paper is well written and organized. However, in my opinion, the research has some flaws regarding some information, and I believe that the same concentration of melatonin is not appropriate for all plant kinds such as cereals, pulses, monocots, dicots, annual plants, and perennial plants. For a better understanding, I propose that melatonin concentrations be described for distinct categories of plants such as cereal crops, pulse crops, oil seed crops, annual plants, perennial plants, and trees. If you apply these suggestions, the report provides useful information for improved understanding. Given these flaws, the manuscript requires major adjustments.

Reviewer 4 Report

This manuscript reports a meta-analysis of the effects of melatonin on plant drought stress alleviation. It provides a general conclusion on the positive contribution of the melatonin on drought stress toleran of plants. Some minor points for revision are given here under:

1. Page 2, line 54; please use the word "plant" in place of "crop"

2. Page 3, Table 1; please use the word "plant height" in place of "plant high"

3. Page 5, Figure 1; please use the word "plant height" in place of "plant high"
